# Frailty prevalence in older adults with atrial fibrillation: A cross-sectional study in a resource-limited setting

José Guillermo Colchado Vallejos[1,2]*, Gonzalo Daniel Quirós López[2]*, Tania Tello Rodríguez[1,2,3], Henry Anchante Hernández[2,3]

**1** Instituto de Gerontología, Universidad Peruana Cayetano Heredia, Lima, Perú, **2** Facultad de Medicina Alberto Hurtado, Universidad Peruana Cayetano Heredia, Lima, Perú, **3** Departamento de Medicina, Hospital Nacional Cayetano Heredia, Lima, Perú

\* jose.colchado.v@upch.pe (JGCV); gonzalo.quiros.l@upch.pe (GDQL)

**Editor:** Paul-Adrian Calburean, George Emil Palade University of Medicine Pharmacy Science and Technology of Targu Mures: Universitatea de Medicina Farmacie Stiinte si Tehnologie George Emil Palade din Targu Mures, ROMANIA

## Abstract

### Background/Objectives

Frailty is a common condition among older adults and is associated with an increased risk of adverse health outcomes, including mortality, disability, dysmobility, falls, and hospitalization. In patients with atrial fibrillation (AF), these risks are further exacerbated. However, evidence linking AF and frailty, particularly in the South American context, is limited. This study aimed to assess frailty and other geriatric conditions in older outpatients with atrial fibrillation in a resource-limited setting in Lima, Peru.

### Methods

In this cross-sectional study, we included adults aged 60 years and older diagnosed with atrial fibrillation who were attending outpatient check-ups. Patients who were hospitalized, receiving chemotherapy induction, or presenting with acute infections or exacerbations were excluded. Standardized questionnaires were used to assess frailty, cognitive impairment, and functional dependence. Statistical analysis was performed using R Studio version 4.3.1, with a significance level set at $p < 0.05$.

### Results

Among the 200 patients who agreed to participate (mean age 74.76 ± 8.42 years, 41% females), 28.5% exhibited frailty, and 46.5% were classified as prefrail. Frailty and prefrailty were significantly associated with older age ($p<0.01$), female gender ($p = 0.01$), illiteracy ($p<0.01$), heart failure ($p<0.01$), falls ($p<0.05$), cognitive impairment ($p<0.01$), and functional dependence ($p<0.01$). Multivariate analysis revealed significant associations between frailty and cognitive impairment ($p<0.05$), frailty and functional dependence ($p<0.05$), and cognitive impairment and functional dependence ($p<0.05$).

**Data Availability Statement:** All relevant data are within the manuscript and its Supporting Information files.

**Funding:** The author(s) received no specific funding for this work.

**Competing interests:** The authors have declared that no competing interests exist.

## Conclusions

One-third of older outpatients with atrial fibrillation were identified as frail, while half were classified as prefrail. In this population, frailty frequently coexists with cognitive impairment and functional dependence, highlighting the need for timely screening and the implementation of evidence-based interventions for individuals with atrial fibrillation in resource-limited settings.

## Introduction

Atrial Fibrillation (AF) stands as the most prevalent arrhythmia among the adult population [1–3]. According to the Global Burden Disease Project, its global prevalence is estimated at 37,574 million cases (4,977 cases per million inhabitants) [1]. Approximately 1 in 4 adults are at risk of developing AF during their lifetime [2]. This risk is influenced by age and the accumulation of various risk factors, including comorbidities, unhealthy lifestyles, genetic predisposition and environmental factors [3]. Frailty and cardiovascular diseases share a bidirectional association [4, 5], driven by common risk factors and underlying pathophysiological mechanisms [5]. In an era of increasing life expectancy, age-related conditions such as frailty and atrial fibrillation come to the forefront [6].

Frailty is a geriatric syndrome defined by a decline in physiological reserves, leading to vulnerability to both internal and external stressors [7, 8]. Two conceptual frameworks have emerged to understand frailty. The phenotypic model, developed by Linda Fried, conceptualizes frailty as the deterioration of an individual's physical performance. Conversely, the multidimensional model, initially proposed by Rockwood and Mitnitsky, defines frailty as the cumulative result of medical, functional and psychosocial deficits, which are integrated into a frailty index [7]. The prevalence of frailty varies by factors including sex, age, setting and the assessment tool used in each study [7, 9]. In a meta-analysis, the overall prevalence of frailty was found to be 12% using the frailty phenotype and 23% for the frailty index [9]. Recent studies indicate that patients with AF exhibit a higher incidence [10] and prevalence of frailty [11, 12]. In fact, a systematic review involving older individuals with AF estimated an overall frailty prevalence of 39.7% [13]. Furthermore, frailty among patients with AF is associated with an elevated risk of all-cause mortality, major bleeding and acute ischemic stroke [6, 13]. AF patients also display a higher prevalence of falls in the past year, polypharmacy, functional dependence and cognitive dysfunction, irrespective of their stroke history [10, 11].

South America is a region marked by significant cultural diversity and economic inequities. The notable increase in life expectancy is paired with poor standards of living and a larger incidence of chronic diseases and disability, in comparison to more developed countries. This complex phenomenon might increase frail populations in coming years [8]. This study assessed frailty status and other geriatric syndromes among older outpatients with atrial fibrillation at a general hospital in Peru.

## Methods

### Study design

This cross-sectional study included patients recruited from the cardiology, geriatrics, and internal medicine outpatient clinics at Cayetano Heredia National Hospital (HNCH) in Lima, Peru, between May 3rd and July 5th, 2023. Longitudinal study designs were not feasible due to

limited resources and personnel. Given the inadequate registration of AF prevalence in the hospital system, a non-probabilistic convenience sampling method was used.

## Participants

Adults aged 60 and older with a documented diagnosis of atrial fibrillation based on conventional electrocardiogram recordings or Holter studies in their medical records were included. Exclusion criteria encompassed patients who were unable to communicate with the interviewers, those with current hospitalization, a history of acute ischemic stroke or myocardial infarction within the last 3 months, individuals undergoing induction chemotherapy for malignant neoplastic disease, those experiencing acute infections, or those dealing with acute exacerbations of chronic illnesses. Prior to participation, eligible patients were invited to provide informed consent. The HNCH is a resource-limited hospital affiliated with the Peruvian Ministry of Health network, primarily providing care to individuals in extreme poverty or with low incomes who are not covered by other insurance plans. Moreover, these individuals reside in the northern districts of Lima, which are among the most densely populated outskirts in the city and have a high rate of aging.

## Variables

Sociodemographic characteristics, comorbidities, medications, and prescribed oral anticoagulants were collected from participants' medical records and personal interviews conducted by the authors. Polypharmacy was defined as the concurrent use of five or more medications, as reported by the participant [14]. Falls syndrome was defined as the history of either two falls or a single fall that required medical attention within the past year. Cognitive impairment was assessed using Pfeiffer's Short Portable Mental Status Questionnaire (SPMSQ), with a threshold of three or more points indicating dysfunction (S1 Fig) [15]. Functional dependence was measured using the Barthel Index, where a score of 100 points denoted independence and a score below 100 points indicated some degree of dependence (S1 Fig) [16]. FRAIL scale was used to assess frailty status, which is already adapted and validated in the Spanish language (S1 Fig) [17]. This scale is a comprehensive tool that encompasses both phenotypic and multidimensional frameworks. It evaluates fatigue (constant or near-constant tiredness), endurance (the ability to climb one flight of stairs without fatigue or stopping), ambulation (the capacity to walk one block without exhaustion or interruption), weight loss (exceeding 5% in the last 6 months), and the presence of more than five diseases from a total list of 11 [4, 18, 19]. Prefrailty is identified with a score of 1–2 points, while frailty is defined by a score of 3–5 points.

## Statistical analyses

For the descriptive analysis, quantitative variables were summarized using measures of central tendency (mean, median) and dispersion (standard deviation, interquartile range) while qualitative variables were presented as absolute numbers and relative frequencies. In the bivariate analysis, variables were examined with respect to frailty status. Cochran-Armitage test, also known as the Chi-square test for trend ($X^2_t$), was employed to compare categorical variables. When dealing with polytomous variables, Chi-square test based on a Monte Carlo simulation ($X^2_{MC}$) was used. Trends in quantitative variables were assessed using the Jonckheere-Terpstra test (JT). To investigate the relationship between frailty status, cognitive impairment and functional dependence, a multivariate approach that included the log-linear model of homogeneous association was employed (S1 Table). Variables were dichotomized to enhance the statistical analysis. A p-value of $< 0.05$ was considered statistically significant. The statistical analysis was performed using the R Studio statistics package, version 4.3.1.

### Ethics

Approval was obtained from the local Institutional Research Ethics Committee (CIEI in Spanish) of the Cayetano Heredia Peruvian University. The recruitment period started on May 3rd and ended on July 5th, 2023. All participants of the study and/or their companions were informed about the research methods, and their consent was obtained by signing the approved written informed consent form.

## Results

A total of 203 patients with a diagnosis of AF were initially included in the study. 200 of them agreed to sign the informed consent. Among these participants, 164 were recruited from cardiology, 21 from geriatrics and 15 from internal medicine outpatient service. The mean age of participants was 74 ± 8 years. Of the participants, 59% were male and 41% were female. Hypertension was the most prevalent comorbidity, affecting 61% of participants, followed by heart failure (37%), dyslipidemia (31.5%) and previous history of acute ischemic stroke (21.5%). Frailty was identified in 28.5% of participants while prefrailty status, in 46.5% (Fig 1). The median score on the FRAIL scale was 1.5, with an interquartile range of 2.75 points. In the bivariate analysis, the robust, prefrail and frail groups showed progressively higher mean age, greater prevalence of female sex, cognitive impairment, falls, illiteracy, and heart failure (Table 1). Multivariate analysis (Fig 2) revealed significant associations between frailty and

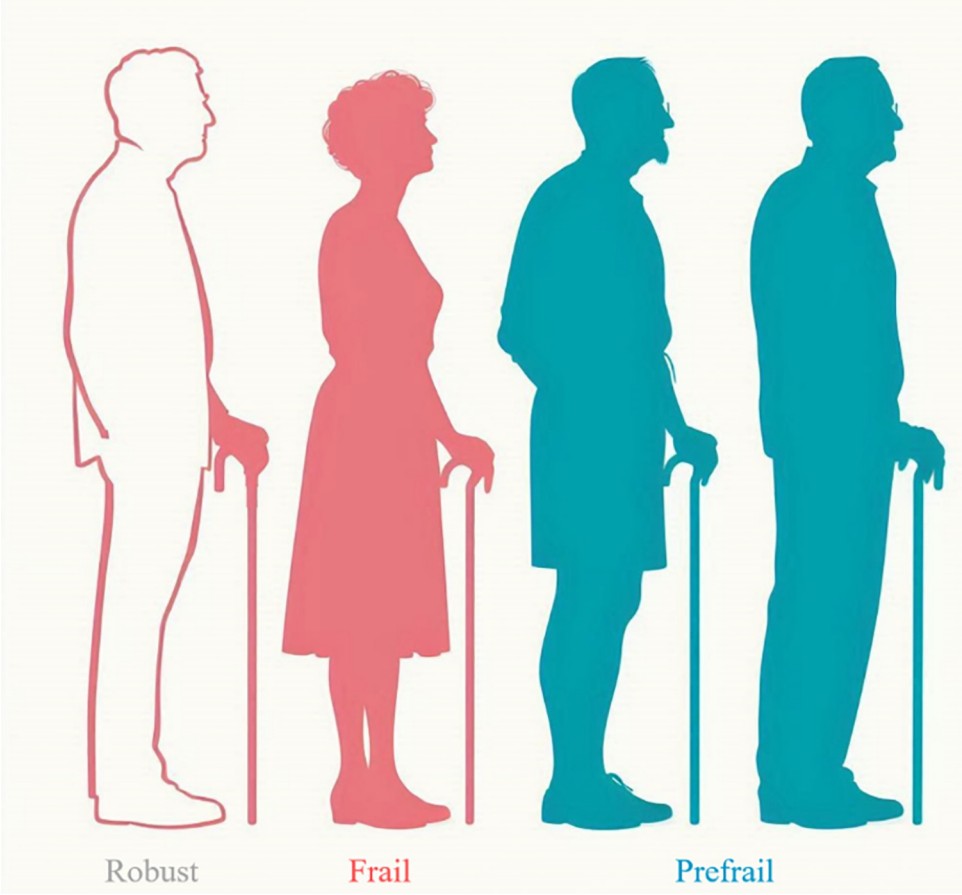

**Fig 1. Frailty prevalence.** In this resource-limited setting, 25% of patients were classified as robust (white), 46.5% as prefrail (blue) and 28.5% as frail (red).

**Table 1. Clinical characteristics of the study population according to frailty status.**

| | Total | Robust | Prefrail | Frail | *p* |
|---|---|---|---|---|---|
| | (n = 200) | (n = 50) | (n = 93) | (n = 57) | |
| | n (%) or M ± SD or m ⇔ IQR | n (%) or M ± SD or m ⇔ IQR | n (%) or M ± SD or m ⇔ IQR | n (%) or M ± SD or m ⇔ IQR | |
| Age (years) | 74.8 ± 8.4 | 71.8 ± 7.2 | 75.3 ± 8.2 | 76.2 ± 9.3 | **0.005** |
| Sex | | | | | |
| Male | 118 (59%) | 37 (74%) | 53 (57%) | 28 (49.1%) | |
| Female | 82 (41%) | 13 (26%) | 40 (43%) | 29 (50.9%) | **0.01** |
| Educational level | | | | | **0.018** |
| Non-educated | 51 (25.5%) | 7 (14%) [+] | 20 (21.5%) [▽] | 24 (42.1%) [+▽] | **0.008** |
| Elementary school | 70 (35%) | 19 (38%) | 31 (33.3%) | 20 (35.1%) | |
| Middle school | 54 (27%) | 17 (34%) | 28 (30.1%) | 9 (15.8%) | |
| College | 25 (12.5%) | 7 (14%) | 14 (15.1%) | 4 (7%) | |
| Comorbidities | | | | | |
| Arterial Hypertension | 122 (61%) | 33 (66%) | 55 (59.1%) | 33 (57.9%) | 0.397 |
| Diabetes Mellitus | 34 (17%) | 6 (12%) | 19 (20.4%) | 9 (15.8%) | 0.641 |
| Chronic Lung Disease | 16 (8%) | 2 (4%) | 8 (8.6%) | 6 (10.5%) | 0.22 |
| Ischemic heart disease | 29 (14.5%) | 9 (18%) | 13 (14%) | 7 (12.3%) | 0.407 |
| Heart failure | 74 (37%) | 9 (18%) | 33 (35.5%) | 32 (56.1%) | < **0.001** |
| History of acute ischemic stroke | 43 (21.5%) | 9 (18%) | 24 (25.8%) | 10 (17.5%) | 0.905 |
| Chronic Kidney Disease | 14 (7%) | 3 (6%) | 6 (6.5%) | 5 (8.8%) | 0.567 |
| Dyslipidemia | 63 (31.5%) | 17 (34%) | 30 (32.3%) | 16 (28.1%) | 0.504 |
| History of tuberculosis | 12 (6%) | 2 (4%) | 6 (6.5%) | 4 (7%) | 0.52 |
| History of iatrogenic bleeding | 33 (16.5%) | 6 (12%) | 16 (17.2%) | 11 (19.3%) | 0.316 |
| Implanted cardiac devices | 15 (7.5%) | 4 (8%) | 7 (7.5%) | 4 (7%) | 0.847 |
| Oral Anticoagulant | | | | | 0.158 |
| VKA | 96 (48%) | 27 (54%) | 47 (50.5%) | 22 (38.6%) | |
| DOACs | 85 (42.5%) | 17 (34%) | 36 (38.7%) | 32 (56.1%) | |
| Apixaban | 66 (33%) | 11 (22%) | 28 (30.1%) | 27 (47.4%) | |
| Rivaroxaban | 15 (7.5%) | 5 (10%) | 5 (5.4%) | 5 (8.8%) | |
| Dabigatran | 4 (2%) | 1 (2%) | 3 (3.2%) | 0 (0%) | |
| None | 19 (9.5%) | 6 (12%) | 10 (10.75%) | 3 (5.3%) | |
| Number of medications | 4 ⇔ 3 | 4 ⇔ 2.25 | 4 ⇔ 3 | 5 ⇔ 3 | 0.104 |
| < 5 drugs | 104 (52%) | 27 (54%) | 49 (52.7%) | 28 (49.1%) | |
| ≥ 5 drugs | 96 (48%) | 23 (46%) | 44 (47.3%) | 29 (50.9%) | |
| Falls [a] | 75 (37.5%) | 12 (24%) | 37 (39.8%) | 26 (45.6%) | **0.023** |
| Cognitive impairment (*Pfeiffer*) | 1 ⇔ 3 | 0 ⇔ 1 | 1 ⇔ 3 | 2 ⇔ 4 | < **0.001** |
| Normal | 142 (71%) | 48 (96%) | 65 (69.9%) | 29 (50.9%) | |
| Cognitive impairment | 58 (29%) | 2 (4%) | 28 (30.1%) | 28 (49.1%) | |
| Functional dependence (*Barthel Index*) | 92.5 ⇔ 15 | 100 ⇔ 5 | 95 ⇔ 15 | 85 ⇔ 15 | **0.001** |
| 100 points | 64 (32%) | 33 (66%) | 26 (28%) | 5 (8.8%) | |
| 90–95 points | 70 (35%) | 16 (32%) | 37 (39.8%) | 17 (29.8%) | |
| < 90 points | 66 (33%) | 1 (2%) | 30 (32.3%) | 35 (61.4%) | |

n = number of participants; % percentage; M: Mean; SD: Standard Deviation; m: Median; IQR: Interquartile range; VKA: Vitamin K antagonist; DOACs: Direct Oral Anticoagulants; [+▽] points in which row p value is < 0.05

[a] In the same year ≥ 2 uncomplicated falls o 1 complicated fall. Bold values indicate a *p*-value < 0.05.

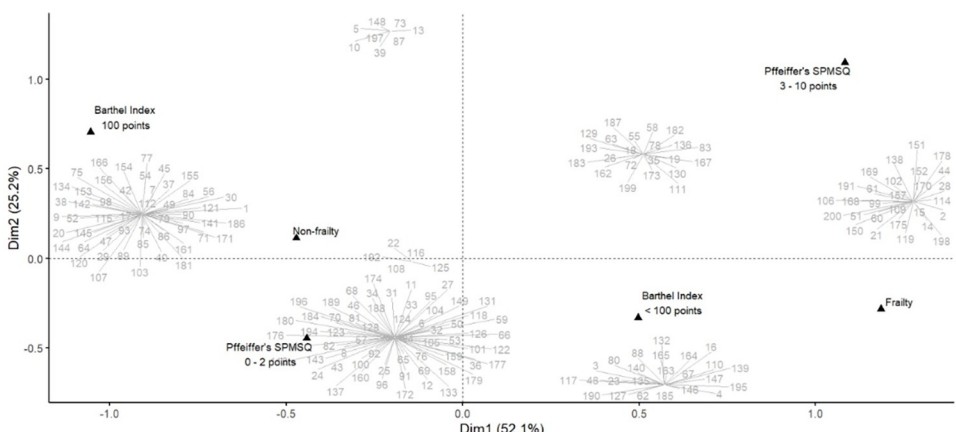

**Fig 2. Two-dimensional correspondence analysis plot of the multivariate analysis.** This graphic visually depicts the close association between non-fragile, non-dependent, and non-cognitive impaired participants. Few participants were exceptions to the rule in the multivariate analysis. Frailty status (FRAIL ≥3), cognitive impairment (Pfeiffer's SPMSQ ≥3), and functional dependency (Barthel Index <100) were dichotomized and represented by black triangles. Each participant is assigned a gray number and grouped in a certain cluster according to their cognitive, frailty and functional status. Shorter graphical distances from the black triangles indicate a stronger statistical association with the corresponding variable. Data loss is minimized by utilizing two primary dimensions (Dim1 and Dim 2). Collectively, these dimensions account for 77.3% of the total variability in the dataset. Moreover, reference values on both the x and y axes, which do not represent physical distances, aid in the graphical distribution of data.

cognitive impairment, irrespective of functionality; frailty and functional dependence, irrespective of cognitive impairment; and between cognitive impairment and functional dependence, regardless of frailty status (Table 2). There was no association among all three variables simultaneously (S2 Table).

## Discussion

This cross-sectional study revealed that 75% of older people diagnosed with atrial fibrillation and attending the outpatient clinic exhibited frailty or prefrailty. The most frequent FRAIL criterion was the incapacity to climb stairs, met by 51% of participants and 96% of those classified as frail, likely due to its demanding physical nature. Remarkably, no patients met all five frailty criteria simultaneously. Similar studies employing the FRAIL scale reported a frailty prevalence from 15.7% to 66.4% [11, 20–25] and prefrailty rates ranging from 40.7 to 51.6% [20–22,

**Table 2. Multivariate analysis. Homogeneous association log-linear model.**

| Coefficients | Estimate | Standard error | Z value | Pr(>\|z\|) | |
|---|---|---|---|---|---|
| (Intercept) | 3.957 | 0.137 | 28.823 | <2e-16 | *** |
| Cognitive impairment | -2.055 | 0.384 | -5.355 | 0.000 | *** |
| Functional dependence | 0.149 | 0.186 | 0.8 | 4.24e-01 | |
| Frailty | -2.649 | 0.477 | -5.559 | 0.0000 | *** |
| Cognitive impairment: Functional dependence | 1.098 | 0.435 | 2.524 | 0.012 | * |
| Cognitive impairment: Frailty | 1.011 | 0.350 | 2.889 | 0.004 | ** |
| Functional dependence: Frailty | 1.774 | 0.507 | 3.502 | 0.0005 | *** |

Frailty: FRAIL scale = 3–5 points; Functional dependence: Barthel index < 100 points; Cognitive impairment: Pfeiffer Questionnaire = 3–10 points

* = $p < 0.05$

** = $p < 0.01$

*** = $p < 0.001$.

24, 25]. This variation can be attributed to geographic and socioeconomic disparities, highly specific inclusion criteria [21, 22], participants with a higher average age [11, 20, 23], inclusion of hospitalized patients [20], the degree of polypharmacy [11, 23, 24] or small sample sizes [24, 25]. Previous publications involving populations with AF have consistently associated frailty with advancing age [13], female sex [13], a lower level of education [21], history of falls in past year [20, 26], a higher number of medications [20], cognitive impairment [20–22, 24] and functional dependence [20]. Multicenter studies with low risk of bias and that recruit patients from the community tend to report lower frailty prevalence rates [13]. This study was conducted at a single center and recruited patients who were undergoing regular check-ups without acute exacerbations, rendering them clinically comparable to individuals in the community with AF. The multivariate analysis suggests that in atrial fibrillation patients, the presence of frailty, cognitive impairment or functional dependence is associated with another of these geriatric syndromes, regardless of the presence of the third. Furthermore, a stronger association was observed between frailty and functional dependence compared to the other pairs of associations evaluated. In patients with AF, cognitive impairment ranged from 32.3% to 35.9%, even when different assessment tools were used [22, 23, 27]. Regarding functional dependence, one study revealed that 39.2% of their participants scored below 60 on the Barthel Index [20], whereas only 6.5% of this study's population fell into this category, maybe due to lower mean age and non-hospitalization status [20]. Other studies examining the link between frailty and functional dependence employed different assessment tools [10, 23] or involved the general population without AF, making direct comparisons unfeasible. In patients with AF, a well-established association exists between frailty and comorbidities such as hypertension, diabetes, ischemic stroke, heart failure and peripheral vascular disease [13]. However, in this research, an association was found only with heart failure.

This population showed a greater percentage of falls syndrome compared to previous studies [20, 23]. This concerning data could be due to a substantial number of older individuals lacking appropriate care and support. In our study, we found 16.5% of self-reported history of iatrogenic bleeding, ranging from dermal hematomas to severe complications. This was not associated with other geriatric variables and may be attributed to a higher risk of trauma in northern Lima districts, which should be confirmed in further studies. Indeed, this neighborhood is known for not being elderly-friendly due to several architectural barriers such as large stairs and hillside houses. In addition, it is important to highlight the elevated prescription rate of oral anticoagulants (90.5%) compared to other researches, which ranged from 61.4% to 72.2% [10, 13, 20–22]. Furthermore, there was no significant association between the use of oral anticoagulants and other geriatric syndromes (S3 Table). It is worth noting that this study did not assess adherence or factors influencing medication access, such as financial constraints. Previous studies have attempted to correlate frailty with lower levels of oral anticoagulant prescription, but they have yielded conflicting results [13, 28, 29]. Physicians may defer anticoagulation prescriptions for various reasons, including active bleeding, prior bleeding episodes, a high risk of major bleeding, dementia, concerns about patient adherence, a history of falls and patient refusal [30]. In this study, no significant differences in reported polypharmacy were observed based on frailty status. This lack of significance could be attributed to potential memory issues among participants or their caregivers, limited awareness of the prescribed medications, or inadequate recording of patient data in medical records.

These findings suggest further screening activities and follow-up for certain geriatric syndromes in AF and frail outpatients. Moreover, our findings justify longitudinal studies in South America. In order to prevent frailty progression and additional resource use in this resource-limited setting, this study suggests comprehensive attention in cardiology and geriatrics services for patients with AF. To our knowledge, this is one of the first studies in South

America that assesses frailty and other geriatric syndromes in patients with atrial fibrillation. Furthermore, patient evaluations were conducted by only two primary researchers, minimizing interobserver variability. However, while the results are representative of northern Lima, they may not be statistically extrapolated to the entire Peruvian population since participants were recruited solely from a single center. Given the cross-sectional design, causality cannot be determined, nor can the evolution over time of frailty status among AF patients. Moreover, non-probabilistic sampling was utilized because there was no accessible and detailed record of AF patients at the host hospital. Additionally, it's plausible that the frailest patients face greater difficulty accessing healthcare services or may be unable to communicate effectively with the research team due to consequences of their illness. This potential limitation should be acknowledged. More resources are needed to design studies that assess anticoagulation effectiveness, prescription and obtain important measurements such as INR.

## Conclusions

Approximately one-third of older outpatients, who were diagnosed with atrial fibrillation and undergoing regular check-ups, were found to be frail, while half of the patients fell into the prefrail category. Furthermore, individuals with atrial fibrillation who exhibit frailty or prefrailty are at an increased risk of experiencing other geriatric syndromes, such as falls within the past year, functional dependence, and cognitive impairment. Given the demographic diversity and geographical variations across South America, our team strongly recommends conducting additional studies on the prevalence of frailty in patients with cardiovascular conditions. For patients with atrial fibrillation, longitudinal studies are needed to better understand the causal relationship with frailty status in South America. Additionally, the high prevalence of frailty observed in our population underscores the importance of timely screening and implementing evidence-based interventions for individuals with atrial fibrillation attending outpatient clinics.

## Supporting information

**S1 Table. Summary of evaluated log-linear models for a 3-way contingency table involving the variables frailty, cognitive impairment, and functional dependence.**
(DOCX)

**S2 Table. Saturation model.** Based on the data (in bold), there is no evidence to support the conclusion that there is a simultaneous interaction between cognitive impairment, frailty, and functional dependence.
(DOCX)

**S3 Table. Oral anticoagulation prescribed according to geriatric syndromes.** Based on the table, there is no statistically significant relation between geriatric syndromes and oral anticoagulation, except for polypharmacy.
(DOCX)

**S1 Fig. Questionnaires used in this study, translated to english by the authors.** (A) FRAIL questionnaire, (B) Pfeiffer Short Portable Mental Status Questionnaire, (C) Barthel Index.
(DOCX)

## Acknowledgments

The authors would like to express their gratitude to José Chauca Carhuajulca, MSc for his expertise in conducting statistical analysis.

## Author Contributions

**Conceptualization:** José Guillermo Colchado Vallejos, Gonzalo Daniel Quirós López.

**Data curation:** José Guillermo Colchado Vallejos, Gonzalo Daniel Quirós López.

**Investigation:** José Guillermo Colchado Vallejos, Gonzalo Daniel Quirós López.

**Methodology:** José Guillermo Colchado Vallejos, Gonzalo Daniel Quirós López.

**Project administration:** José Guillermo Colchado Vallejos, Gonzalo Daniel Quirós López, Tania Tello Rodríguez, Henry Anchante Hernández.

**Resources:** José Guillermo Colchado Vallejos, Gonzalo Daniel Quirós López, Tania Tello Rodríguez, Henry Anchante Hernández.

**Software:** José Guillermo Colchado Vallejos, Gonzalo Daniel Quirós López.

**Supervision:** Tania Tello Rodríguez, Henry Anchante Hernández.

**Validation:** José Guillermo Colchado Vallejos, Gonzalo Daniel Quirós López.

**Visualization:** José Guillermo Colchado Vallejos, Gonzalo Daniel Quirós López.

**Writing – original draft:** José Guillermo Colchado Vallejos, Gonzalo Daniel Quirós López.

**Writing – review & editing:** Tania Tello Rodríguez, Henry Anchante Hernández.

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
