## [Decision Letter · Decision Letter 0]

28 Aug 2024

PONE-D-24-28720Frailty prevalence in Older adults with Atrial fibrillation: a cross-sectional study in a resource-limited settingPLOS ONE

Dear Dr. Colchado Vallejos,

Thank you for submitting your manuscript to PLOS ONE. After careful consideration, we feel that it has merit but does not fully meet PLOS ONE’s publication criteria as it currently stands. Therefore, we invite you to submit a revised version of the manuscript that addresses the points raised during the review process.

We look forward to receiving your revised manuscript.

Kind regards,

Paul-Adrian Calburean

Academic Editor

PLOS ONE

Journal Requirements:

Reviewers' comments:

Reviewer's Responses to Questions

**Comments to the Author**

1. Is the manuscript technically sound, and do the data support the conclusions?

Reviewer #1: Yes

Reviewer #2: Yes

2. Has the statistical analysis been performed appropriately and rigorously? 

Reviewer #1: I Don't Know

Reviewer #2: Yes

3. Have the authors made all data underlying the findings in their manuscript fully available?

Reviewer #1: Yes

Reviewer #2: Yes

4. Is the manuscript presented in an intelligible fashion and written in standard English?

Reviewer #1: Yes

Reviewer #2: Yes

5. Review Comments to the Author

Reviewer #1: There are a few minor revisions that I’m recommending:

1-In the abstract I would recommend elaborating a bit on the definition of frailty in the context of the research. Eg, Frailty is the state where older people who are at highest risk of adverse outcomes such as falls, disability, admission to hospital, or the need for long-term care.

2-In the abstract also; lines 9-12; instead of naming the questionnaires it is possible to mention that you are using standardized screening questionnaires for evaluation of frailty, cognition and functional dependence . Then mention the three questionnaires.

3- A more detailed explanation on what the Frail score, SPMSQ, and Barthes index ask about, or add a copy of each tot he table section.

4- inclusion of more tables and charts to further visualize our findings. Visualization using charts makes the deliverance of information and findings much easer.

5-Identify limitations and dress them earlier in the paper.

This paper is among the first to address Frailty in Af pts in tha geographical area. Further study of the issue is definitely needed.

Thank you for starting the discussion.

Reviewer #2: Well presented study. Frailty is a common factor to be studied among elder people on the field of Cardiology.

Correlation with atrial fibrillation, one of the most common arrhythmia of elderly people is crucial and has clinical impact

Given that both situation affect clinical decisions and management.

6. PLOS authors have the option to publish the peer review history of their article (what does this mean?). If published, this will include your full peer review and any attached files.

Reviewer #1: **Yes: **Maha Ahmed

Reviewer #2: **Yes: **Afendoulis Dimitrios

---

## [Author Response · Author response to Decision Letter 0]

20 Sep 2024

Dear Editorial Team of PLOS ONE,

Dear Reviewers,

We made the requested corrections in order to proceed in our submission. 

1. In the abstract, we added information about the influence of frailty in older adults and simplified the description of the questionnaires used in this study.

2. In the study design section, we addressed some of the study's limitations, including inadequate data registration in the hospital and the limited resources and personnel available for data collection.

3. As per your suggestion, we have included an image that visually represents the prevalence of frailty, prefrailty, and robust participants. We believe this visual representation enhances the clarity and impact of the findings in this resource-limited setting.

4. In the Supporting Information, we have included images of the translated versions of the questionnaires used in this study: the FRAIL scale, the Pfeiffer Short Portable Mental Status Questionnaire, and the Barthel Index.

Modifications are highlighted in yellow in the marked-up copy of our manuscript, while corrections to academic writing are highlighted in blue.

Thank you for your corrections,

José Guillermo Colchado Vallejos, MD

---

## [Editor Report · Decision Letter 1]

8 Oct 2024

Frailty prevalence in Older adults with Atrial fibrillation: a cross-sectional study in a resource-limited setting

PONE-D-24-28720R1

Dear Dr. Vallejos

We’re pleased to inform you that your manuscript has been judged scientifically suitable for publication and will be formally accepted for publication once it meets all outstanding technical requirements.

Kind regards,

Paul-Adrian Calburean

Academic Editor

PLOS ONE

---

## [Editor Report · Acceptance letter]

14 Oct 2024

PONE-D-24-28720R1 

PLOS ONE

Dear Dr. Colchado Vallejos, 

I'm pleased to inform you that your manuscript has been deemed suitable for publication in PLOS ONE. Congratulations! Your manuscript is now being handed over to our production team.

Kind regards, 

on behalf of

Dr. Paul-Adrian Calburean 

Academic Editor

PLOS ONE